# Group-Fair Online Allocation in Continuous Time

**Semih Cayci**[*]
scayci@illinois.edu

**Swati Gupta**[†]
swatig@gatech.edu

**Atilla Eryilmaz**[‡]
eryilmaz.2@osu.edu

## Abstract

The theory of discrete-time online learning has been successfully applied in many problems that involve sequential decision-making under uncertainty. However, in many applications including contractual hiring in online freelancing platforms and server allocation in cloud computing systems, the outcome of each action is observed only after a random and action-dependent time. Furthermore, as a consequence of certain ethical and economic concerns, the controller may impose deadlines on the completion of each task, and require fairness across different groups in the allocation of total time budget $B$. In order to address these applications, we consider continuous-time online learning problem with fairness considerations, and present a novel framework based on continuous-time utility maximization. We show that this formulation recovers reward-maximizing, max-min fair and proportionally fair allocation rules across different groups as special cases. We characterize the optimal offline policy, which allocates the total time between different actions in an optimally fair way (as defined by the utility function), and impose deadlines to maximize time-efficiency. In the absence of any statistical knowledge, we propose a novel online learning algorithm based on dual ascent optimization for time averages, and prove that it achieves $\widetilde{O}(B^{-1/2})$ regret bound.

## 1   Introduction

With the prevalence of automated decision methods and machine learning methods, it is important to analyze the impact of learning and evaluate models not only with respect to traditional objectives such as reward or model accuracy, but also to account for the impact on individuals that interact with the system. Indeed, there are many studies highlighting algorithmic discrimination due to problems in the machine learning pipeline: imbalance in data [1], learnt representations [2, 3], choice of model proxies [4], demographic group-dependent difference in error rates of the learned models [5, 6, 7], to name a few. With rising ethical and legal concerns, addressing such issues has become urgent, specially as these impact critical societal decisions involving job opportunities and hiring. In 2014, it was estimated that 25% of the total workforce in the US was involved in some form of freelancing, and this number was predicted to grow to 40% by 2020 [8]. In reality, this percentage might be much higher, due to COVID-19 restrictions leading to increased work-from-home and changes in job opportunities [9, 10]. In online platforms however, there has been a strong evidence of bias observed in number of user reviews and user ratings[4] on completing jobs with significant correlations

---

[*]Coordinated Science Laboratory, University of Illinois at Urbana-Champaign, Urbana, IL 61801

[†]H. Milton Stewart School of Industrial and Systems Engineering, Georgia Institute of Technology, Atlanta, GA 30332

[‡]Department of Electrical and Computer Engineering, The Ohio State University, Columbus, OH 43210

[4]The mean (median) normalized rating score for White workers was 0.98 (1), while it is 0.97 (1) for Black workers on TASKRABBIT. The mean (median) rating of White workers was found to be 3.3 (4.8), 3.0 (4.6) for Black workers, 3.3 (4.8) for Asian workers, 3.6 (4.8) for workers with a picture that does not depict a person, and 1.7 (0.0) for workers with no image on FIVERR [11].

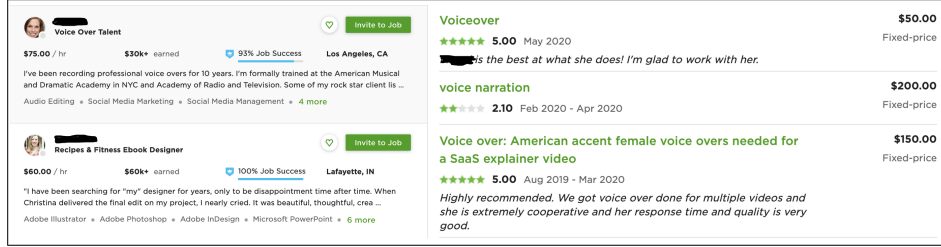

Figure 1: Freelancer profiles on UPWORK with their past performance and corresponding reviews for "fixed-price" contracts. Contractors can access these profiles and allocate fixed-timed contracts with deadlines.

with race, gender, location of work and length of profiles[5] [11]. Motivated by these problems in online contractual hiring, we study a theoretical framework for sequential resource allocation to workers, where the controller (decision maker) can enforce deadlines for each task's completion. Our key contribution is to quantify impact of reward maximization in terms of equality of opportunity for jobs and develop algorithms that can achieve a meaningful trade-off between these via online utility maximization. The challenge is to maximize total reward within a given time budget, while accounting for random completion times by workers from different groups and fairness in allocation.

Formally, we consider $K$ groups of individuals who can be hired sequentially for each task, i.e., at any point, exactly one individual can be hired. If an individual from group $k \in [K]$ is chosen for the $n$-th task and given a contractual deadline $t$ by the controller, he/she generates a random *reward* of $R_{k,n}$ if the task is completed by (random) time $X_{k,n}$ within deadline $t$. If the task is not completed by the deadline, the reward obtained by the controller is zero and the time until the deadline is wasted (i.e., yields 0 reward for the controller). Completion times and reward distributions are assumed group-dependent and i.i.d. across tasks. The objective of the controller is to maximize utility (trade-off between total reward and fair allocation) in the offline (known distributions) and online settings (unknown distributions) under a budget constraint on time. As we will show in this paper, controlled deadlines set are essential for optimal time-efficiency under the budget constraint.

The ethical problems we are concerned with involve the rate of jobs allocated to different demographic groups and the deadlines imposed on these under reward maximization regimes [11]. Our sequential framework would also apply to other settings, for e.g., comparative clinical trials with varying follow-up durations as well as to server allocation in cloud computing where jobs are drawn from different application groups and must commit computational resources until a specific amount of time due to service level agreements (Section 2). We will often focus on the first application involving online contractual hiring, since fairness concerns are most naturally motivated in this domain.

Given a time budget constraint $B$ and the diverse random nature of completion time and reward pairs, the main question we consider is how to decide distribution of tasks and deadlines between different groups of people. Two potential extreme allocations are: (i) *Reward-maximizing task allocation*: The controller assigns all tasks to the most rewarding group to maximize the total reward within the given time budget. The other groups do not get any chance to receive tasks. (ii) *Proportional task allocation*: The controller completely ignores the reward distributions, and attempts to give equal time share to each group. In other words, each group receives a fraction of the tasks inversely proportional to their mean completion times. There is clearly a trade-off between the reward maximization and equal time-share considerations in continuous-time sequential task allocation, and well-chosen utility functions [12] can be helpful in modeling this in a unified way. In this paper, we consider a very general class of utility functions, which recovers broadly used fairness criteria such as proportional fairness, max-min fairness, reward maximization among many others [13, 14, 12]. The controller can determine her priorities in terms of notions of fairness and model the task allocation problem by choosing the utility function accordingly.

The main contributions of this paper are summarized as follows:

1. **Incorporation of random completion time dynamics and fairness in allocation:** In discrete-time online learning models, each action is assumed to take a unit completion time, thus the

random and diverse nature of task completion times, as required in many fundamental real-life applications, is ignored. In this work, we incorporate this aspect and develop a sequential learning framework in continuous time using tools from the theory of renewal processes and stochastic control. We show how controlled deadlines improve the time-efficiency in continuous-time decision processes. Moreover, this is the first work, to the best of our knowledge, that analyzes fair distribution policies in online contractual hiring.

2. **Characterization of Approximately Optimal Offline Policies:** As a consequence of the random and controlled task completion times, the optimal policy for fair resource allocation is PSPACE-hard akin to unbounded stochastic knapsack problems. For tractability in design and analysis, we propose an approximation to the optimal offline policy based on Lagrange duality and renewal theory, and prove that it is asymptotically optimal. These approximate policies allocate tasks independently with respect to a fixed probability distribution.

3. **Online learning for utility maximization:** For utility maximization in an online setting with full information feedback, we develop a novel and low-computational-complexity online learning algorithm based on dynamic stochastic optimization methods for time averages, and show that it achieves $\tilde{O}(B^{-1/2})$ regret for a time budget $B$. The optimal offline control policy in this paper is time-dependent, randomized and attempts to optimize time averages unlike the reward maximization problems in discrete-time problems. Despite these, the online learning algorithm we developed adapts to the randomness in completion time-reward pairs, and achieves optimal performance with vanishing regret at a fast rate.

**Related Work:** The problem of fair resource allocation via utility maximization has been widely considered in economics and network management [15, 16, 17, 18]. The utility maximization approach to fair resource allocation in these papers predominantly deals with discrete-time systems, therefore the randomness and diversity in task completion times is completely ignored. Furthermore, these works either assume perfect knowledge of rewards and completion times prior to decision-making, or they assume the knowledge of statistics, therefore they do not incorporate online learning. The only continuous-time utility maximization approach to fair resource allocation is [19], which assumes the knowledge of first-order statistics, thus considers an offline optimization setting. Our work utilizes the (offline) Lyapunov optimization methods proposed in [19] to develop online learning algorithms based on the notion of approximate Lyapunov drift.

Online learning under budget constraints has been considered under the scope of bandits with knapsacks [20, 21, 22]. In the classical bandits with knapsacks model, the objective is to maximize expected total reward under knapsack constraints in a stochastic setting. In [23], an interrupt mechanism is employed to incorporate the continuous-time dynamics into the budget-constrained online learning model. Note that these works focus solely on reward maximization, therefore do not address the fair resource allocation problem. The bandits with knapsacks setting was extended to concave rewards and convex constraints in [24], which assumes bounded cost and reward, and the deadline mechanism is not involved in decision-making, thus optimal time-efficiency in continuous time is not achieved. Our paper deviates from this line of work as it proposes a versatile and comprehensive framework for fairness, and incorporates continuous-time dynamics into the decision-making for time-efficiency. We include an extended discussion of related work in Appendix A.

## 2 Online Learning Framework for Group Fairness

We consider the sequential and fair allocation of tasks to individuals from different groups, whose completion times and rewards randomly vary. This goal differs significantly from traditional online learning models that aim to maximize the expected total reward with unit completion times. Under this traditional setting, the controller's goal is to find and persistently select the reward-maximizing groups to allocate its tasks. As a consequence, the reward-maximization objective leads to the starvation of suboptimal groups, which causes unfairness amongst the groups with different statistical characteristics. Next, we provide a few motivating examples with group fairness requirements:

- **Contractual Hiring in Online Freelancing Platforms:** Online freelancing sites like UPWORK host contractual workers (freelancers) that can be hired by "contractors" who require specific tasks to be completed. Each freelancer has a profile and performance on past tasks that can be learned by the contractors via ratings and reviews (see, typical profile in Figure 1). Fixed-timed contracts are popular on UPWORK, wherein contractors enforce a deadline by which the task must be completed

otherwise the contract is terminated (i.e., there is no payment). Contractors can browse profiles and post a job to a selected set of freelancers with a deadline. However, there is a large literature documenting bias in online rating systems, which in turn impact job opportunities disparately [11, 25, 26], thus making it critical to develop theory of online learning for such settings.

- **Server Allocation in Cloud Computing:** An important application of our framework is online learning for fair resource allocation in cloud computing systems. In a very basic setting, a single server is sequentially allocated to tasks from one of $K$ user groups, which exhibit similar execution time statistics and priority levels within each group. In many practical scenarios, the execution time of a task is unknown at the time decision [27, 28], and exhibits a power-law behavior [29], which necessitates a deadline mechanism for optimal time-efficiency [23]. In this setting, the objective of the controller is to allocate the server in an optimally fair way across the groups in a given time interval $[0, B]$, depending on the completion time statistics and priority levels. Our work proposes a versatile framework to model fairness for this problem based on the concept of continuous-time utility maximization, and develops online learning algorithms to achieve the optimal performance with low regret in the absence of any statistical knowledge.

More examples can be found in other domains, including multi-user wireless communication over fading channels (e.g., see [23]), comparative clinical trials with optimal follow-up duration (e.g., see [30, 31]), whereby the goal is to fairly share the limited resources between groups of users.

Motivated by these examples, next we introduce an online learning framework that expands the traditional setting substantially to incorporate group fairness characteristics into its formulation. Suppose that there are $K \geq 1$ groups of individuals that are available for serving tasks, and the controller assigns tasks sequentially among these $K$ groups. If an individual from group $k$ is chosen for the $n$-th task, he/she takes $X_{k,n}$ units of *completion time*, and a *reward* of $\overline{R}_{k,n}$ is obtained upon successful completion, where $X_{k,n}$ and $\overline{R}_{k,n}$ are positive random variables. We assume that individuals within a group exhibit statistical similarities as a consequence of their common background, therefore we assume that the process $(X_{k,n}, \overline{R}_{k,n})$ is independent and identically distributed (iid) over $n$. In order to model the possibility of highly different skill sets for individuals within a group, we assume that the completion time $X_{k,n}$ is random, and can be potentially heavy-tailed. Note that the completion time $X_{k,n}$ and reward $\overline{R}_{k,n}$ can be correlated, e.g., in the server allocation example, the completion time $X_{k,n}$ and size $\overline{R}_{k,n}$ of a task are positively correlated [32].

Before the $n$-th task begins, the controller makes two decisions: the group $G_n \in [K]$ of the individual that will be assigned the task, and a deadline $T_n \in \mathbb{T}$, where $\mathbb{T} \subset \mathbb{R}_+$ is the set of all possible deadlines. Since $(X_{k,n}, \overline{R}_{k,n})$ is independent and identically distributed over $n$ and unknown at the time of decision, i.e., each individual is statistically symmetric, we do not consider the assignment of tasks to individuals within a group, thus the controller only makes a choice for the group in task allocation. If the task is not completed by the selected deadline $t \in \mathbb{T}$, the task is interrupted without collecting any reward, therefore we denote the reward obtained $t$ time units after the initiation by $R_{k,n}(t) = \overline{R}_{k,n}\mathbb{I}\{X_{k,n} \leq t\} \leq R_{max}$ for some constant $R_{max} < \infty$. In many applications, deadlines are chosen within a discrete set (e.g., days/months in contractual hiring or time-slots in server allocation), thus we assume a finite decision set $\mathbb{T} = \{t_1, t_2, \ldots, t_L\}$ with $t_l < \infty$ for all $l$ in this paper. The sequential task allocation continues until a given time budget $B > 0$ is exceeded, therefore, the completion time of a task is as important as the reward.

To describe this process mathematically, let $\mathcal{H}_{k,n-1}$ denote the available feedback for group $k$, and $\mathcal{H}_{n-1} = \cup_{k \in [K]} \mathcal{H}_{k,n-1}$ denote the history before making a decision for task $n$. For a given time budget $B > 0$, a *causal policy* $\pi = \{\pi_1, \pi_2, \ldots\}$ sequentially makes two decisions $\pi_n = (G_n, T_n) \in [K] \times \mathbb{T}$ for each task $n$ based on the history $\mathcal{H}_{k-1}$, where $G_n$ is the chosen group and $T_n$ is the assigned deadline. Under a policy $\pi$, the number of initiated tasks is the following *first-passage time*:

$$N^\pi(B) = \inf\left\{n : \sum_{i=1}^n \min\{X_{G_i,i}, T_i\} > B\right\}, \tag{1}$$

which is a random and controlled stopping time. Moreover, the *reward rate* of any user type $k$ is:

$$\overline{r}_k^\pi(B) = \mathbb{E}\left[\frac{1}{B} \sum_{n=1}^{N_\pi(B)} \mathbb{I}\{G_n = k\} R_{k,n}(T_n)\right], \text{ under policy } \pi. \tag{2}$$

If $R_{k,n}(t) = \mathbb{I}\{X_{k,n} \le t\}$, i.e., each task completion yields a unit reward, then $\bar{r}_k^\pi(B)$ simply denotes the task completion rate (i.e., throughput) of group $k$ individuals in the time interval $[0, B]$.

Note that designing strategies that aim to maximize the total reward rate in (2) will lead to the persistent selection of the group with the highest reward rate at the cost of starvation of all the rest (see [23]). In order to address group fairness considerations, we propose a continuous-time online learning framework based on the utility maximization concept that is used effectively in the fair resource allocation domain (e.g., see [16]). Specifically, for a given continuously-differentiable, concave and monotonically increasing utility function $U_k : \mathbb{R} \to \mathbb{R}$, we let the utility of group $k$ under a policy $\pi$ be given by $U_k\big(\bar{r}_k^\pi(B)\big)$. Then, the total utility under a policy $\pi$ is defined as:

$$U^\pi(B) = \sum_{k=1}^K U_k\big(\bar{r}_k^\pi(B)\big), \text{ for time interval } [0, B].$$

For a given time budget $B > 0$, the optimum total utility over a class of policies $\Pi$, and the regret for a given policy $\pi \in \Pi$ are, respectively:

$$\texttt{OPT}_\Pi(B) = \max_{\pi \in \Pi} \sum_{k=1}^K U_k\big(\bar{r}_k^\pi(B)\big) \quad \text{and} \quad \texttt{REG}_\Pi^\pi(B) = \texttt{OPT}_\Pi(B) - U^\pi(B). \tag{3}$$

Note that, due to the monotonically increasing and concave nature of utility functions, allocating the tasks always to the most rewarding group is not a good choice, because the same amount of time could yield a higher utility for another group because of the diminishing return property of concave functions. Thus, each given set of utility functions $\{U_k : k \in [K]\}$ defines a fairness concept. A particularly important class of utility functions is $\alpha$-fair class, given next.

**Definition 1** ($\alpha$-Fair Allocation). *For any given $\alpha > 0$ and weight $w_k > 0$, let $U_k(x) = w_k \frac{x^{1-\alpha}}{1-\alpha}$, for all $k$. Resource allocation by using these utility functions is called $\alpha$-fair resource allocation.*

This class is attractive since it includes as special cases proportional fairness, minimum potential delay fairness, reward maximization and max-min fairness [12].

## 3 Approximation of the Optimal Offline Policy

Note that a simpler version of the sequential maximization problem in (3) with linear utility functions over all causal policies is called an unbounded knapsack problem, and it is PSPACE-hard even in the case of known statistics [33, 20]. Therefore, the optimal causal policy for the problem in (3) has a very high computational complexity even in the offline setting, which makes it intractable for online learning. For tractability in design and analysis, we consider a class of simple policies that allocate tasks in an i.i.d. randomized way according to a fixed probability distribution over groups, and show its efficiency in this section.

**Definition 2** (Stationary Randomized Policies). *Let $P$ be a fixed probability distribution over $[K] \times \mathbb{T}$. A stationary randomized policy (SRP) $\pi = \pi(P)$ makes a randomized decision independently according to $P$ for every task until the time budget $B$ is depleted. In other words, under the SRP $\pi(P)$, we have $\mathbb{P}\big(\pi_n = (k, t)\big) = P(k, t), \forall n \le N_\pi(B),$ for all $(k, t) \in [K] \times \mathbb{T}$. We denote the class of all stationary randomized policies as $\Pi_S$.*

**Proposition 1** (Asymptotic optimality of SRP). *There exists a probability distribution $P^\star$ such that the stationary randomized policy $\pi(P^\star)$ is asymptotically optimal over all causal policies as $B \to \infty$.*

The proof of Proposition 1 can be found in Appendix B. In the following, we characterize the total utility under $\pi(P)$ by providing tight bounds.

**Proposition 2.** *Let $P$ be any given probability distribution over $[K] \times \mathbb{T}$. Then, the reward per unit time for group $k$ under the stationary randomized policy $\pi(P)$ is as follows:*

$$\rho_k(P) = \frac{\sum_{t \in \mathbb{T}} P(k,t)\mathbb{E}[R_{k,1}(t)]}{\sum_{(i,t) \in [K] \times \mathbb{T}} P(i,t)\mathbb{E}[\min\{X_{i,1}, t\}]}, \forall k \in [K].$$

*Consequently, the total utility under the stationary randomized policy $\pi(P)$ is bounded as follows:*

$$\sum_{k \in [K]} U_k\Big(\rho_k(P)\Big) \le \sum_{k \in [K]} U_k\big(\bar{r}_k^{\pi(P)}(B)\big) \le \sum_{k \in [K]} U_k\Big(\rho_k(P)\Big) + O\Big(\frac{1}{B}\Big).$$

We include the complete proof of Proposition 2 in Appendix B. The key idea is that under an SRP, the total reward of a group $k$ is a regenerative process. Then, by using the theory of stopped random walks for regenerative processes, the reward per unit time under $\pi(P)$ is found as $\rho_k(P)$, and the upper bound for the total utility is found by using Lorden's inequality [34] and concavity of $U_k$.

Proposition 2 emphasizes the significance of the reward per unit time $\rho_k(P)$. In conjunction with Proposition 1, this suggests that using a probability distribution that maximizes the limiting total utility would be an effective offline approximation.

**Definition 3** (Optimal Stationary Randomized Policy). *Let $P^\star$ be a probability distribution defined as $P^\star \in \arg\max_P \sum_{k \in [K]} U_k\Big(\rho_k(P)\Big)$. Then, the optimal SRP $\pi^\star$ makes a selection independently for every task according to $P^\star$: $\mathbb{P}\Big(\pi_n^\star = (k, t)\Big) = P^\star(k, t)$ for all $(k, t) \in [K] \times \mathbb{T}$ and $n \le N_\pi(B)$.*

An interesting question regarding $P^\star$ is the choice of deadline policy for each group. The following proposition characterizes the optimal deadline policy under $\pi^\star$, and yields a significant simplification in finding the optimal policy by reducing the size of the search space.

**Proposition 3** (Optimal Deadline Policy). *For any $k$, the optimal probability distribution $P^\star$ makes a deterministic deadline decision for group $k$, that is, $|\{t \in \mathbb{T} : P^\star(k, t) > 0\}| \le 1$. For any $k$, we denote $t_k^* \in \mathbb{T}$ as the (unique) optimal deadline for group $k$ such that $P^\star(k, t_k^*) > 0$.*

The detailed proof of Prop. 3 can be found in Appendix C. As we will see later, we can explicitly characterize the optimal deadline for a broad class of utility functions used for the so-called $\alpha$-fair allocations. In the following, we use Prop. 2 to characterize the performance of the optimal SRP.

**Proposition 4** (Optimal Total Utility). *For any group $k$, let $t_k^* \in \mathbb{T}$ be the (unique) optimal deadline by Prop. 3; $r_k^* = \mathbb{E}[R_{k,1}(t_k^*)]/\mathbb{E}[\min\{X_{k,1}, t_k^*\}]$ be the reward per processing time for group $k$; and*

$$\varphi_k = \frac{P^\star(k, t_k^*) \cdot \mathbb{E}[\min\{X_{k,1}, t_k^*\}]}{\sum_{j \in [K]} P^\star(j, t_j^*) \cdot \mathbb{E}[\min\{X_{j,1}, t_j^*\}]}, \tag{4}$$

*be the fraction of time budget allocated to group $k$ under $\pi(P^\star)$. Then, for any SRP $\pi(P)$, the total utility is bounded as $\sum_k U_k\Big(\rho_k(P)\Big) \le \sum_k U_k\Big((U_k')^{-1}\big(\frac{\lambda}{r_k^*}\big)\Big)$, where the upper bound is achieved by the probability distribution that satisfies $\varphi_k = \frac{1}{r_k^*}(U_k')^{-1}\big(\frac{\lambda}{r_k^*}\big)$ for $\lambda$ such that $\sum_k \varphi_k = 1$.*

The proof of Proposition 4 follows from Lagrange duality and Prop. 3, and can be found in Appendix D. Note that the above analysis is very general in the sense that it holds for any set of utility functions $\{U_k : k \in [K]\}$ that are continuously differentiable and concave. In the following, we apply the results to the class of $\alpha$-fair allocations (cf. Definition 1) and discuss their implications.

**Proposition 5** ($\alpha$-Fair Resource Allocation in Continuous Time). *For any group $k$, the optimal deadline optimizes the reward per processing time:*

$$t_k^* = \arg\max_{t \in \mathbb{T}} \frac{\mathbb{E}[R_{k,1}(t)]}{\mathbb{E}[\min\{X_{k,1}, t\}]}.$$

*Let $r_k^* = \max_{t \in \mathbb{T}} \frac{\mathbb{E}[R_{k,1}(t)]}{\mathbb{E}[\min\{X_{k,1}, t\}]}$ be the reward per processing time and $\mu_k = \mathbb{E}[\min\{X_{k,1}, t_k^*\}]$ be the mean processing time for group $k$ under the optimal deadline policy. Then, for any $\alpha > 0$, we have the following results for $\alpha$-fair utility functions:*

$$\max_P U^{\pi(P)}(B) = \frac{1}{1 - \alpha}\Big( \sum_{k \in [K]} (r_k^*)^{\frac{1}{\alpha} - 1} w_k^{\frac{1}{\alpha}} \Big)^\alpha, \tag{5}$$

*where the optimum probability distribution $P_k^*$ and the optimum fraction of time budget $\varphi_k$ allocated to group $k$ are, respectively, given by:*

$$P^\star(k,t) = \mathbb{I}\{t = t_k^*\} \frac{w_k^{\frac{1}{\alpha}} (r_k^*)^{\frac{1}{\alpha}-1}/\mu_k}{\sum_{j \in [K]} w_j^{\frac{1}{\alpha}} (r_j^*)^{\frac{1}{\alpha}-1}/\mu_j},$$

$$\varphi_k = \frac{(r_k^*)^{\frac{1}{\alpha}-1} w_k^{\frac{1}{\alpha}}}{\sum_{j \in [K]} (r_j^*)^{\frac{1}{\alpha}-1} w_j^{\frac{1}{\alpha}}},$$

*for all $(k,t) \in [K] \times \mathbb{T}$.*

Proposition 5 characterizes the optimal stationary randomized policy for a wide class of utility functions, and implies that the optimal deadline for a group is chosen to maximize the reward per processing time for that group in this setting.

To gain a clear understanding of the notion of $\alpha$-fairness, we consider the following special cases.

**Corollary 1.** *For any given set of parameters $\{w_k > 0 : k \in [K]\}$, we have the following results for continuous-time $\alpha$-fair resource allocation problem for various $\alpha > 0$ values.*

(i) ***Proportional fairness:*** *In this case, we have $\lim_{\alpha \to 1} U_k(x) = w_k \log(x)$ for all $k$. Let $\mu_k = \mathbb{E}[\min\{X_{k,1}, t_k^*\}]$ be the mean processing time for group $k$. Then, the optimum utility is achieved by the probability distribution $P^\star(k,t) = \mathbb{I}\{t = t_k^*\} \frac{w_k/\mu_k}{\sum_{j \in [K]} w_j/\mu_j}$, $(k,t) \in [K] \times \mathbb{T}$, thus we have $\varphi_k = \frac{w_k}{\sum_{j \in [K]} w_j}$ for all $k$, which implies the time budget $B$ is allocated to a group $k$ proportional to its weight $w_k$, and the optimal total utility is $\mathtt{OPT}_{\Pi_S}(B) = \sum_k \log\left(\frac{r_k^* w_k}{\sum_{k' \in [K]} w_{k'}}\right) + O(\frac{1}{B})$.*

(ii) ***Reward maximization:*** *If $\alpha = 0$, we have $U_k(x) = \omega_k x$ for all $k$. Let $k^* = \arg\max_{k \in [K]} w_k r_k^*$ be the group with highest weighted reward rate. Then, the optimal probability distribution is $P^\star(k,t) = \mathbb{I}\{k = k^*, t = t_k^*\}$, for all $(k,t)$. Thus, $\mathtt{OPT}_{\Pi_S}(B) = \max_{k \in [K]} w_k r_k^* + O(1/B)$.*

**Remark 1.** Note that optimal deadline $t_k^*$ for any group $k$ is chosen so as to maximize the reward per processing time of group $k$. Under proportional fairness ($\alpha \to 1$), the controller distributes the time budget proportional to group weights, i.e., $\varphi_k = w_k / \sum_j w_j$, which reduces to equal time-sharing under uniform weights. To achieve this, the controller allocates tasks with probability inversely proportional to the mean processing time $\mu_k$. Under reward maximization ($\alpha = 0$), the controller allocates the entire time budget $B$ to a single group that yields the highest reward per processing time to maximize the expected total reward, i.e., $\varphi_k = \mathbb{I}\{k = k^*\}$. As such, the trade-off between reward maximization and equal (i.e., reward-insensitive) time-sharing is modeled by $\alpha$-fairness for any $\alpha \in [0, 1)$. Further, the $\alpha$-fair utility maximization framework includes max-min fairness ($\alpha \to \infty$) and minimum potential delay fairness ($\alpha = 2$) as subcases.

## 4  Online Learning for Utility Maximization (OLUM)

In the previous section, we provided key results on the asymptotically optimal approximations to the offline utility maximization problem. In this section, we will build on these to attack the online learning problem for continuous-time fair allocation. In particular, we will propose a novel online learning algorithm for the fair resource allocation problem based on renewal theory, Lyapunov optimization (see [19, 35]) and PAC-bandits, and show that it achieves $\widetilde{O}(B^{-1/2})$ regret.

**Feedback model:** We assume a delayed full-information feedback model where the completion time and reward of all groups for task $n$ are revealed to the controller at stage $n + \tau$ for some delay $\tau \geq 1$.

This assumption holds approximately for our target applications. In freelancing platforms, there are often multiple contractors that hire freelancers for various tasks. It is often possible to get full information on various freelancers due to employment by other companies and their reviews can serve as the feedback for the controller. Competitions hosting websites like TOPCODER have also recently been catering to businesses who need fast-prototyping using freelancers. In their business model, a controller might invest in a few topcoders at a time, however, she can potentially get access to updated rankings (quality and time to complete tasks) via topcoder competitions over time. In

server applications such as Amazon AWS and Microsoft Azure as well, although a controller might be optimizing operations on a local set of servers, they can request task performance data from a centralized server or a scheduler after a delay in time [36]. This feedback model already presents with technical challenges due to random completion times, as we discuss next.

In order to design the online learning algorithm, let us define, for any $(k,t) \in [K] \times \mathbb{T}$, the empirical estimates of the mean completion time and reward after $n$ stages, respectively, as

$$\widehat{\mu}_{k,n}(t) \ = \ \frac{1}{n} \sum_{i=1}^{n} \min\{t, X_{k,i}\}, \qquad \text{and} \quad \widehat{\theta}_{k,n}(t) \ = \ \frac{1}{n} \sum_{i=1}^{n} R_{k,i}(t).$$

**Definition 4** (OLUM Algorithm). *For any $k$, let $Q_{k,0} = 1$ and $Q_{k,i}$ be defined recursively as follows:*

$$Q_{k,i+1} = \Big( Q_{k,i} + \gamma_k(i) \min\{X_{G_i,i}, T_i\} - R_{k,i}(T_i) \mathbb{I}\{G_i = k\} \Big)^+, \qquad i > 0 \tag{6}$$

*where the auxiliary variable $\gamma_k(i) = (U'_k)^{-1}\Big(Q_{k,i}/V\Big)$, where $V > 0$ is a design choice. Then, for the task $n$, the OLUM Algorithm, denoted by $\pi^{\text{OLUM}}$, makes the following decision:*

$$(G_n, T_n) \in \underset{(k,t)\in[K]\times\mathbb{T}}{\arg\max} \frac{\widehat{\theta}_{k,n-\tau}(t) Q_{k,n}}{\widehat{\mu}_{k,n-\tau}(t)}.$$

*Upon observing the corresponding feedback, the controller updates $Q_{k,n+1}$ via (6).*

**Interpretation:** The OLUM Algorithm aims to maximize the time-average reward weighted with $Q_{k,n}$ at each round. Note that for any $k \in [K]$, if the sequence $Q_{k,n}$ gets very big, then its reward rate is much smaller than the optimal value, thus the controller tends to select that group. In other words, the magnitude of $Q_{k,n}$ is a measure of the unfairness that group $k$ has endured by stage $n$. The algorithm is designed so as to balance the weights $Q_{k,n}$ to maximize the total utility.

In the following theorem, we prove regret bounds for the OLUM Algorithm.

**Theorem 1** (Regret bounds for OLUM). *For any $V > 0$ and constant delay $\tau$, the regret under $\pi^{\text{OLUM}}$ is bounded as $\text{REG}_{\Pi_S}{}^{\pi^{\text{OLUM}}}(B) = O\Big(\sqrt{\frac{\log(B)}{B}} + \frac{V}{B} + \frac{1}{V}\Big)$. By choosing $V = \Theta(\sqrt{B/\log(B)})$, we obtain $\text{REG}_{\Pi_S}{}^{\pi^{\text{OLUM}}}(B) = O(\sqrt{\log(B)/B}) = \widetilde{O}(1/\sqrt{B})$.*

The proof is based on renewal theory, Lyapunov optimization theory and PAC-type bounds, and can be found in Appendix E.

## 5   Simulations

We implemented the OLUM Algorithm on a fair resource allocation problem with $K = 2$ groups. In the application domains that we considered in Section 2, the task completion times naturally follow a power-law distribution. For the contractual online hiring setting, creativity of individuals has been shown to follow a Pareto$(1, \gamma)$ distribution with exponent $\gamma > 1$, where $\gamma$ is dependent on the field of expertise [37]. Motivated by this application, we consider the following group statistics:

- **Group 1:** $X_{k,n} \sim \text{Pareto}(1, 1.2)$ and $R_{k,n}(t) = X_{k,n}^{0.6} \cdot \mathbb{I}\{X_{k,n} \leq t\}$
- **Group 2:** $X_{k,n} \sim \text{Pareto}(1, 1.4)$ and $R_{k,n}(t) = X_{k,n}^{0.2} \cdot \mathbb{I}\{X_{k,n} \leq t\}$

The reward per processing time as a function of the deadline is shown in Figure 2. For this setting, we implemented the OLUM Algorithm with parameter $V = 20$, and considered $\alpha$-fair resource allocation problems with various $\alpha$ values. In Figure 2, we present the simulation results for $\varphi_2$, i.e., the average fraction of time budget $B$ allocated to Group-2 individuals, under the OLUM Algorithm. For these experiments, we chose $w_k = 1$ for $k = 1, 2$ and ran the OLUM Algorithm for 1000 trials for each set. Note that the optimal reward per processing time of Group-1 individuals is higher than that of Group-2 individuals, thus Group-1 is chosen for reward maximization. Under proportional fairness, the time budget is equally distributed between Group-1 and Group-2 individuals. We observe from Figure 2 that the OLUM Algorithm converges to the optimal operating points very fast, which verifies the theoretical results we presented.

In the second example, we consider a fair server allocation problem with $K = 5$ user groups. In processing systems with shared servers, empirical studies indicate that the distribution of job

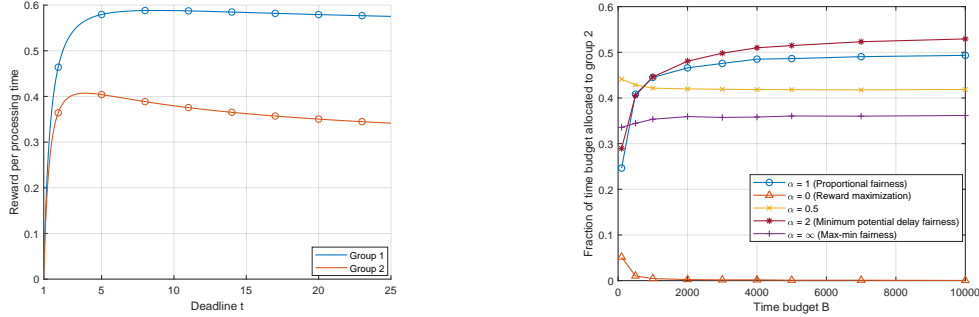

Figure 2: (Left) Reward per processing time for each group. (Right) Fraction of time budget assigned to Group-2 individuals under the `OLUM` Algorithm for various fairness criteria.

execution times (i.e., file sizes) can be accurately approximated by Pareto($1, \gamma$) distribution with $\gamma \in (0, 2)$ [38, 39, 40]. Thus, we consider Pareto distributed completion times for each group with varying exponents: $X_{k,n} \sim Pareto(1, \alpha_k)$ for $\alpha = (1.25, 1.50, 1.75, 2.00, 2.25)$. The reward of group $k$ under a deadline $t \in \mathbb{T}$ is $R_{k,n}(t) = \rho_k \cdot X_{k,n}^{\beta_k} \cdot \mathbb{I}\{X_{k,n} \leq t\} \leq \rho_k t_L^{\beta_k}$ where $\beta_k$ measures the correlation. For this example, we consider $\beta = (0, 0.125, 0.25, 0.375, 0.5)$ and $\rho = (3.0, 2.0, 3.0, 1.0, 1.44)$. Reward per processing times as a function of deadline are shown in Fig. 3, where the deadline choices $t \in \mathbb{T}$ are marked by squares. We observe that the optimal deadline critically depends on the tail exponent and correlation between completion time and reward [23]. The performance of the `OLUM` Algorithm for proportionally fair allocation of the server (i.e., $U_k(x) = \log(x)$, $k \in [K]$) is shown in Fig. 3 for various choices of the parameter $V$. In Theorem 1, we showed that $\widetilde{O}(1/\sqrt{B})$ regret is achieved when the parameter $V$ is chosen as $V = \Theta(\sqrt{B/\log(B)})$ for time budget $B$. From Fig. 3, we verify that this scaling of the parameter $V$ is necessary for good practical performance.

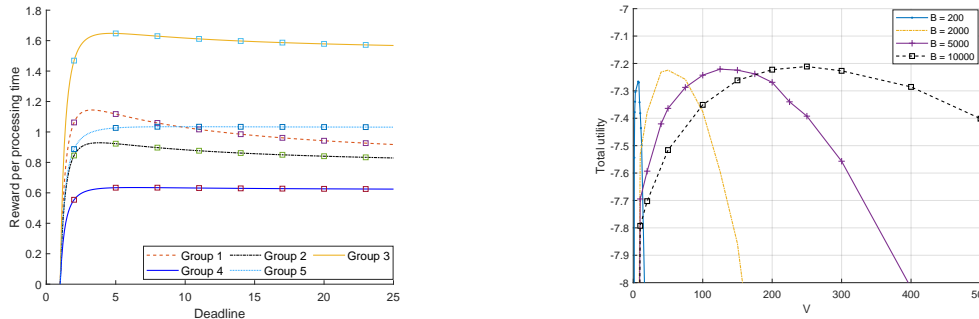

Figure 3: (Left) Reward per processing time for each group. (Right) Total utility for various time budgets and design parameter $V$ choices.

## 6    Conclusion

In this paper, we proposed a versatile and comprehensive framework for continuous-time online resource allocation with fairness considerations, and proposed a no-regret learning algorithm for this problem in a delayed full-information feedback model. Note that although the full-information feedback is available in many application scenarios, there are cases in which the controller does not have an access to full feedback, thus a mechanism that incorporates bandit feedback is required. The online learning framework introduced in this paper can be extended to bandit feedback. One way to achieve this might be to replace the empirical estimates with upper confidence bounds in the `OLUM` Algorithm, which makes the analysis even more complicated. We leave the design and analysis of bandit algorithms in this setting as a future work.

## Broader Impact

Our work develops the theory of fair online learning, specifically analyzing the impact of reward-maximizing allocation policies on opportunities for different groups of people. Our proposal analyzes the trade-offs across various allocation policies (ranging from profit maximizing to equal opportunity for all), thus highlighting the choice of objectives that the controllers should carefully consider. This work does not have any foreseeable negative ethical or societal impact.

## Acknowledgments and Disclosure of Funding

This work was completed when S. Cayci was a PhD candidate and research assistant at The Ohio State University, Department of Electrical and Computer Engineering. This research was supported in part by: NSF grants: CNS-NeTS-1717045, CMMI-SMOR-1562065, CNS-ICN-WEN-1719371, CNS-SpecEES-1824337, CNS-NeTS-2007231; ONR Grant N00014-19-1-2621, and the DTRA grant: HDTRA1-18-1-0050. S. Gupta would like to gratefully acknowledge support from the NSF grant CRII 1850182.

## Footnotes

[5]Mean (median) number of reviews: for women 33 (11), 59 (15) for men on TASKRABBIT. Mean (median) number of reviews: for Black workers was found to be 65 (4), 104 (6) for White workers, 101 (8) for Asian workers, 94 (10) for non-human pictures and 18 (0) for users with no image on FIVERR [11].

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
