[Supplementary Material]

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

 studied in economics [41, 42, 18], mechanism design [43], network management [15, 44, 16, 12, 45] among many other fields. Particularly, logarithmic utility maximization was introduced in [41] for the "Nash bargaining solution" to a bargaining game among multiple players over the allocation of a shared resource, and it was used in the management of communication networks in [44]. As a unifying framework, the class of $\alpha$-fair (also known as "isoelastic") utility functions was proposed for fair allocation in economics in [42]. The main methodology for fair resource allocation in time-varying dynamical systems, akin to the system considered here, is Lyapunov drift analysis. Lyapunov drift has been used as a fundamental design and analysis tool in many problems including the wireless scheduling problem [46, 47], fair resource allocation among competing users [16, 12], stochastic game theory [48]. Based on Lyapunov-drift methods, stochastic dynamic optimization algorithms by using the so-called drift-plus-penalty method were widely used in queueing and networking problems (see [35] and references therein). The existing Lyapunov optimization methods are predominantly opportunistic, which means that the random quantities (such as completion time, reward, system state) arrive prior to the decision-making at each stage, or they assume the knowledge of the first- and second-order statistics of these random quantities. These assumptions are not satisfied in many applications as we discussed in Section 1, therefore the controller must learn the statistics so as to maximize the objective function, such as the total utility. To the best of our knowledge, our paper is the first learning theory approach to the fair resource allocation problem based on Lyapunov drift. Even in the offline optimization setting, the Lyapunov optimization methods are predominantly in discrete-time setting, i.e., each action takes a unit time. The only continuous-time utility maximization approach to fair resource allocation is [19], which assumes the knowledge of first-order statistics. Our work develops an online optimization algorithm based on the Lyapunov optimization methodology proposed in [19] for offline optimization (i.e., when the first-order moments are known), by improving some of the results (e.g., simplified decision rules, finite-time performance bounds), and adapting these for the online learning problem.

The online learning problem under budget constraints has been considered in the bandits with knapsacks (BwK) framework [20]. In this extension of the classical stochastic bandit model, each action consumes a random amount of a resource from a common budget and yields a random reward, where the controller aims to maximize the expected total reward by until a resource is completely depleted. BwK model has been considered under various dynamics [20, 49, 50, 21]. In [23], an interrupt/deadline mechanism is employed to incorporate the continuous-time dynamics into the budget-constrained online learning model. For a detailed discussion of the BwK and its extensions, please refer to [22]. The original BwK models study the reward maximization problem. In [24], the authors consider an online learning setting where the objective is to maximize a concave function subject to convex constraints. In [24], the decision-making process continues for a *fixed* number of stages, and the constraints are not always satisfied unlike our model. Instead, the distance to the constraint set, as well as the regret, is shown to vanish in expectation under the proposed learning algorithms, which require solving linear programs at each stage. Another crucial difference is that the deadline mechanism for improving time-efficiency is not incorporated into the decision in [24]. Our paper deviates from this line of work as it proposes a versatile and comprehensive framework for fairness, and incorporates continuous-time dynamics into the decision-making for time-efficiency under strict time constraints. To solve this problem, we propose a learning algorithm with low computational complexity, and prove its efficiency. The design and analysis methodology we followed in this paper based on Lyapunov optimization can be used in many other problem models.

## B  Proofs of Proposition 1 and Proposition 2

*Proof of Proposition 2.* Fix any (group, deadline) decision $(k, t) \in [K] \times \mathbb{T}$, and consider the stationary policy $\pi = \pi(P)$ with an arbitrary probability distribution $P$. Let the number of $(k, t)$ decisions in $[0, B]$ be defined as

$$N_\pi^{(k,t)}(B) = \sum_{n=1}^{N_{\pi(P)}(B)} \mathbb{I}\{\pi_n = (k, t)\}.$$

Since each decision is made independently according to the same probability distribution, the number of tasks between two consecutive tasks for which the decision is $(k, t)$ is iid, which implies that $N_\pi^{(k,t)}(B)$ is a regenerative process [34]. Therefore, we can compute the total reward gathered from tasks for which the decision-pair is $(k, t)$ by using renewal theory. In order to accomplish this, we will compute the mean length of a regenerative cycle for each decision $(k, t)$, and then use the renewal theory for tight bounds.

Without loss of generality, consider a regenerative cycle from the beginning (time 0) to the completion of the first task where the decision-pair is $(k, t)$, thus each regenerative cycle contains exactly one task for which the decision-pair is $(k, t)$. Then, for the random variable $M = \sup\{n \geq 0 : \pi_n(P) \neq (k, t)\}$, the number of tasks in a regenerative cycle is $M + 1 \sim Geo(P(k, t))$. This construction implies that $\{M = 0\} = \{\pi_1(P) = (k, t)\}$ and $\{M = m\} = \bigcap_{i=1}^{m}\{\pi_i(P) \neq (k, t)\} \cap \{\pi_{M+1}(P) = (k, t)\}$ for $m > 1$ under the stationary randomized policy $\pi(P)$. Therefore, the length of the regenerative cycle (i.e., the time interval in which there is exactly one completed task with decision-pair $(k, t)$) is as follows:

$$Y = \sum_{n=1}^{M} \sum_{(k',t')\neq(k,t)} \mathbb{I}\{\pi_n = (k', t')\}(X_{k',n} \wedge t') + (X_{k,M} \wedge t),$$

where $x \wedge y = \min\{x, y\}$ for any two real numbers $x, y$. Note that $Y$ is a stopped random walk with non-i.i.d. increments and a controlled stopping time $M + 1$. We will compute the expectation of this quantity first. By iterated expectation, we have the following equality:

$$\mathbb{E}[Y] = \sum_{n_0=0}^{\infty} \mathbb{P}(M = n_0)\mathbb{E}[Y|M = n_0]. \tag{7}$$

Note that for any $n_0 \geq 0$, we have:

$$\mathbb{E}[\mathbb{I}\{\pi_n = (k', t')\}|M = n_0] = \mathbb{P}(\pi_n = (k', t')|\pi_n \neq (k, t)) = \frac{P(k', t')}{1 - P(k, t)},$$

for all $n \leq n_0$. Therefore, we have the following identity:

$$\mathbb{E}[Y|M = n_0] = n_0 \sum_{(k',t')\neq(k,t)} \frac{P(k', t')\mu(k', t')}{1 - P(k, t)} + \mu(k, t), \quad \forall n_0 \leq 0, \tag{8}$$

where $\mu(k, t) = \mathbb{E}[X_{k,1} \wedge t]$. Thus, we have the following:

$$\mathbb{E}[Y] = \sum_{n_0=0}^{\infty} \mathbb{P}(M = n_0)n_0 \sum_{(k',t')\neq(k,t)} \frac{p(k', t')\mu(k', t')}{1 - p(k, t)} + \mu(k, t),$$

$$= \mathbb{E}[M] \sum_{(k',t')\neq(k,t)} \frac{p(k', t')\mu(k', t')}{1 - p(k, t)} + \mu(k, t).$$

from (7). Since $M + 1 \sim Geo(P(k, t))$, we have $\mathbb{E}[M] = \frac{1}{p(k,t)} - 1$. Substituting this into the above identity, we find the expected length of a regenerative cycle under $\pi(P)$ as follows:

$$\mathbb{E}[Y] = \frac{\sum_{k',t')\neq(k,t)} P(k', t')\mu(k', t')}{P(k, t)}.$$

In summary, a decision-pair $(k, t)$ is chosen once in a cycle of $Y$ time units, and yields a reward $R_{k,n}(t)$ under the stationary randomized policy $\pi(P)$. Having specified mean length of a regenerative cycle and mean reward, we can now compute the reward rate (i.e., reward per unit time) for a decision-pair $(k, t)$ under $\pi(P)$ as follows:

$$r_k(t) = \frac{\mathbb{E}[R_{k,1}(t)]}{\mathbb{E}[Y]} = \frac{P(k, t)\mathbb{E}[R_{k,1}(t)]}{\sum_{(i,t')\in[K]\times\mathbb{T}} P(i, t')\mathbb{E}[\min\{X_{i,1}, t'\}]}.$$

As an immediate consequence, the reward per unit time for group $k$ under $\pi(P)$ is as follows:

$$\rho_k(P) = \sum_{t\in\mathbb{T}} r_k(t).$$

As a consequence of the elementary renewal theorem [51], the total reward for group $k$ under $\pi(P)$ in $[0, B]$ is $B\rho_k(P) + o(B)$. In order to get tight bounds, we use Lorden's inequality to obtain the following inequalities:

$$B\rho_k(P) \leq \sum_{n=1}^{N_\pi(B)} \sum_{t \in \mathbb{T}} \mathbb{I}\{\pi_n = (k, t)\} R_{k,n}(t) \leq B\rho_k(P) + C(k, t),$$

for a constant $C(k, t) < \infty$ since $Var(X_{k,n} \wedge t) < \infty$ and $Var(R_{k,n}(t)) < \infty$ for all $t \in \mathbb{T}$ [34]. Therefore,

$$\rho_k(P) \leq \overline{r}_k^\pi(B) \leq \rho_k(P) + \frac{C(k, t)}{B}.$$

Since $U_k$ is continuously differentiable and concave, we have the following result:

$$U_k(\rho_k(P)) \leq U_k(\overline{r}_k^\pi(B)) \leq U_k(\rho_k(P)) + U_k'\Big(\rho_k(P)\Big)\frac{C(k, t)}{B},$$

which concludes the proof. □

*Proof of Proposition 1.* First, we will show an approximation to the optimization problem in (3) based on Jensen's inequality.

**Lemma 1.** *For any $k \in [K]$, $n \geq 1$ and a causal policy $\pi$ for choosing $(G_n, T_n, (\gamma_{k,n})_{k \in [K]})$, let*

$$\widetilde{X}_{\pi_n,n} = \min\{X_{G_n,n}, T_n\}, \tag{9}$$

$$Z_{\pi_n,n} = \min\{X_{G_n,n}, T_n\} \sum_{m=1}^K U_m(\gamma_{m,n}), \tag{10}$$

$$Y_{\pi_n,m,n} = \min\{X_{G_n,n}, T_n\}\gamma_{m,n} - R_{m,n}(T_n)\mathbb{I}\{G_n = m\}, \quad \forall m \in [K]. \tag{11}$$

*Let $U^*$ be the solution to the following optimization problem:*

$$\max_{\pi \in \Pi_A} \lim_{N \to \infty} \frac{\sum_{n=1}^N \mathbb{E}[Z_{\pi_n,n}]}{\sum_{n=1}^N \mathbb{E}[\widetilde{X}_{\pi_n,n}]} \quad s.t. \quad \lim_{N \to \infty} \frac{\sum_{n=1}^N \mathbb{E}[Y_{\pi_n,m,n}]}{\sum_{n=1}^N \mathbb{E}[\widetilde{X}_{\pi_n,n}]} \leq 0, \; \forall m = 1, 2, \ldots, K. \tag{12}$$

*where the maximization is over $\Pi_A$, the set of all causal policies. Then, we have the following result:*

$$\lim_{B \to \infty} \text{OPT}_{\Pi_A}(B) = U^*.$$

*Proof.* First, under any policy $\pi \in \Pi_A$, the following holds by the definition of $N^\pi(B)$:

$$\sum_{n=1}^{N^\pi(B)-1} \widetilde{X}_{\pi_n,n} < B \leq \sum_{n=1}^{N^\pi(B)} \widetilde{X}_{\pi_n,n}.$$

Since $\widetilde{X}_{\pi_n,n}$ is bounded for all $n$, we have the following:

$$\lim_{B \to \infty} \overline{r}_k^\pi(B) = \lim_{B \to \infty} \frac{\mathbb{E}\Big[\sum_{n=1}^{N_\pi(B)} \mathbb{I}\{G_n = k\} R_{k,n}(T_n)\Big]}{\mathbb{E}\Big[\sum_{n=1}^{N_\pi(B)} \widetilde{X}_{\pi_n,n}\Big]} \tag{13}$$

By using the asymptotic equality in (13), continuity of $U_k$, and a direct application of the extended Jensen's inequality (see Lemma 7.6 in [35]), we have $\lim_{B \to \infty} \text{OPT}_{\Pi_A}(B) = U^*$. This enables us to convert the utility maximization problem into a constrained optimization for time-averages. □

Now, we will prove the following:

$$\max_P \sum_{k \in [K]} U_k\big(\rho_k(P)\big) = U^*.$$

Since $U^*$ is optimal asymptotic total utility over $\Pi_A \supset \Pi_S$, we have the following inequality:

$$\max_P \sum_{k \in [K]} U_k\big(\rho_k(P)\big) \leq U^*.$$

By using (13), a direct application of Lemma 1 in [19] implies that there exists an SRP $\pi(P_0)$ that achieves $U^*$. Proposition 2 implies that

$$\sum_k U_k(\bar{r}_k^{\pi(P')}(B)) \leq \max_P \sum_k U_k(\rho_k(P)) + O(1/B),$$

for any $P'$ and $B > 0$. Thus, we have:

$$U^* = \lim_{B \to \infty} \sum_k U_k\left(\bar{r}_k^{\pi(P_0)}(B)\right) \leq \max_P \sum_k U_k(\rho_k(P)),$$

which implies $U^* = \max_P \sum_k U_k(\rho_k(P))$. $\qquad\qquad\square$

## C  Proof of Proposition 3

Let $\mu(k,t) = \mathbb{E}[\min\{X_{k,1}, t\}]$ and $\theta(k,t) = \mathbb{E}[R_{k,1}(t)]$. For the optimal distribution $P^\star$, let

$$C_k = \sum_{k' \neq k} \sum_{t \in \mathbb{T}} P^\star(k',t)\mu(k',t),$$

$P_k^\star = [P^\star(k,t)]_{t \in \mathbb{T}}$ and $p_k = \sum_{t \in \mathbb{T}} P^\star(k,t)$. Then, since $U_k(x)$ is an increasing function of $x$, $P_k^\star$ is the solution to the following optimization problem:

$$\max_{P_k} \ \frac{\sum_t P_k(t)\theta(k,t)}{\sum_t P_k(t)\mu(k,t) + C_k} \quad \text{subject to} \quad P_k(t) \geq 0, \forall t,$$
$$\sum_t P_k(t) = p_k. \tag{14}$$

Let $V^*$ be the optimum solution of (14), and $V(P_k) = \sum_t P_k(t)\theta(k,t) - V^*\left(\sum_t P_k(t)\mu(k,t) + C_k\right)$. Then, the following optimization problem is equivalent to (3):

$$\max_{P_k} \ V(P_k) \quad \text{subject to} \quad P_k(t) \geq 0, \forall t,$$
$$\sum_t P_k(t) = p_k, \tag{15}$$

which, in turn, yields $P_k^\star$. For any $t \in \mathbb{T}$, we have $\frac{\partial V}{\partial P_k(t)} = \theta(k,t) - V^*\mu(k,t)$. Let

$$d^* = \max_t \left.\frac{\partial V(P_k)}{\partial P_k(t)}\right|_{P_k = P_k^\star}.$$

By the optimality of $P_k^\star$, if $P_k^\star(t) > 0$, then we must have $\partial V(P_k^\star)/\partial P_k(t) = d^*$, which further implies that

$$P_k^\star(t) > 0 \Rightarrow \theta(k,t) = d^* + V^*\mu(k,t). \tag{16}$$

Let $t_1 \leq t_2 \leq \ldots \leq t_m$ be the set of deadlines such that $P_k^\star(t_i) > 0$. There exists a $\beta \in [0,1]$ such that the following holds:

$$\sum_t P_k^\star(t)\theta(k,t) = p_k\left(\beta\theta(k,t_1) + (1-\beta)\theta(k,t_m)\right).$$

In conjunction with (16), this implies that:

$$\sum_t P_k^\star(t)\mu(k,t) = p_k\left(\beta\mu(k,t_1) + (1-\beta)\mu(k,t_m)\right).$$

Hence, we have shown that $P_k^\star$ makes a randomization between at most two deadlines, which simplifies (15) considerably as a function of a single variable $\beta \in [0,1]$. Rewriting (15) in terms of $\beta$ and taking the derivative with respect to $\beta \in [0,1]$, we observe that the objective function is either monotonically decreasing or increasing with $\beta$. Therefore, $P_k^\star$ has only one non-zero element, i.e., the deadline decision is made deterministically for group $k$.

# D Proof of Proposition 4

By Proposition 3, for each group $k$, there is a unique optimal deadline $t_k^*$. Let

$$r_k^* = \frac{\mathbb{E}[R_{k,n}(t_k^*)]}{\mathbb{E}[\min\{X_{k,n}, t_k^*\}]},$$

be the reward per processing time for group $k$ under the optimal deadline selection. Then, by Proposition 2, we can express the reward per unit time as follows:

$$\rho_k(P) = r_k^* \widehat{\varphi}_k(P),$$

where

$$\widehat{\varphi}_k(P) = \frac{P(k, t_k^*)\mathbb{E}[\min\{X_{k,n}, t_k^*\}]}{\sum_{j \in [K]} P(j, t_j^*)\mathbb{E}[\min\{X_{j,n}, t_j^*\}]},$$

is the fraction of time allocated to group $k$ under $\pi(P)$. Note that for any $P$, $\{\widehat{\varphi}_k(P) : k \in [K]\}$ defines a probability distribution in the $K$-dimensional simplex. Therefore, by Proposition 2, the asymptotically optimal utility is the solution to the following optimization problem:

$$\max_{\varphi \in \mathbb{R}_+^K} \sum_{k \in [K]} U_k(r_k^* \widehat{\varphi}_k) \quad \text{s.t.} \quad \sum_{k \in [K]} \widehat{\varphi}_k = 1, \tag{17}$$
$$\widehat{\varphi}_k \geq 0, \ \forall k \in [K].$$

The Lagrangian function associated with (17) is as follows:

$$\mathcal{L}(\widehat{\varphi}, \lambda) = \sum_{k \in [K]} U_k(r_k^* \widehat{\varphi}_k) - \lambda\Big(\sum_{k \in [K]} \widehat{\varphi}_k - 1\Big).$$

Since $U_k$ is a monotonically increasing and continuously differentiable function for all $k$, by solving $\frac{\partial \mathcal{L}}{\partial \widehat{\varphi}_k} = 0$, we obtain $\widehat{\varphi}_k = (U_k')^{-1}(\lambda/r_k^*)$. As $U_k$ is concave for all $k$, the proof follows by applying KKT conditions.

# E Proof of Theorem 1

The proof of Theorem 1 consists of two steps. In the first step, we analyze the performance of the OLUM Algorithm for the constrained optimization of time averages for any number of trials $N$ by using a Lyapunov optimization methodology [19, 35, 48]. In the second step, we show that the number of tasks processed in $[0, B]$ is $O(B)$ with high probability to prove the regret result.

The following concentration inequality will be used extensively throughout the proof.

**Lemma 2** ([52, 23]). *Let $X_n$ and $R_n$ be two sub-Gaussian random processes with means $\mathbb{E}[X] > 0$, $\mathbb{E}[R]$, and parameters $\sigma_X^2$ and $\sigma_R^2$, respectively. Then, for any $\epsilon \in (0, \mathbb{E}[X])$, we have the following:*

$$\mathbb{P}\Big(\Big|\frac{\sum_{i=1}^n R_i}{\sum_{i=1}^n X_i} - \frac{E[R]}{E[X]}\Big| > \frac{\epsilon(1+r)}{\mu}\Big) \leq 2\big(e^{-n\epsilon^2/\sigma_X^2} + e^{-n\epsilon^2/\sigma_R^2}\big), \tag{18}$$

*for any $r > \frac{\mathbb{E}[R]}{\mathbb{E}[X]}$ and $\mu \leq \mathbb{E}[X] - \epsilon$.*

Note that any bounded random variable $Z \in [0, a]$ is sub-Gaussian with parameter $\sigma^2 = a^2/4$ [53]. As we are dealing with bounded $\min\{X_{k,n}, t\}$ and $R_{k,n}(t)$, Lemma 2 is an essential result for the proofs in this section.

In the second lemma, we provide an upper bound for the expectation of the dual variables $Q_n = (Q_{1,n}, Q_{2,n}, \ldots, Q_{K,n})$.

**Lemma 3.** *Consider the dual variables defined in (6) under the OLUM Algorithm, and without loss of generality assume $Q_{k,0} = 1$ for all $k$. Then, we have the following bound for any $n \geq 1$:*

$$\mathbb{E}[\sum_{k=1}^K Q_{k,n}] \leq V \sum_{k \in [K]} U_k'\Big(\frac{\min_{k,t} \mathbb{E}[R_{k,n}(t)] - \epsilon}{\max_{k \in [K]} \mathbb{E}[X_{k,n}]}\Big) + O(1/\epsilon), \tag{19}$$

*for any $V > 0$ and $\epsilon \in \big(0, \min_{k,t} \mathbb{E}[R_{k,n}(t)]\big)$.*

*Proof.* For any $\epsilon \in \big(0, \min_{k,t} \mathbb{E}[R_{k,n}(t)]\big)$, let

$$A = \{q : \sum_k q_k \geq V \sum_{k \in [K]} U_k'\Big(\frac{\min_{k,t} \mathbb{E}[R_{k,n}(t)] - \epsilon}{\max_{k \in [K]} \mathbb{E}[X_{k,n}]}\Big) + \max_t \overline{R}_{max}(t)\}.$$

Then, we have $\mathbb{E}[\sum_k Q_{k,n+1} - \sum_k Q_{k,n}; Q_n \in A \big| Q_n] \leq -\epsilon$. Also, note that $Q_{k,n+1} - Q_{k,n}$ is bounded almost surely, i.e., sub-Gaussian. Thus, Theorem 2.3 in [54] implies the tail bounds for $\sum_k Q_{k,n}$, which implies the result via $\mathbb{E}[X\mathbb{I}\{X > a\}] = a\mathbb{P}(X > a) + \int_a^\infty \mathbb{P}(X > x)dx$. $\qquad \square$

**Step 1:** Recall the equivalent form of the utility maximization problem in Lemma 1. In this step, we will prove the following result under the `OLUM` Algorithm:

$$\frac{\mathbb{E}[\sum_{n=1}^N Z_{\pi_n,n}]}{\mathbb{E}[\sum_{n=1}^N \widetilde{X}_{\pi_n,n}]} \geq U^* - O\Big(\sqrt{\frac{\log(N)}{N}} + \frac{1}{V}\Big),$$

$$\frac{\mathbb{E}[\sum_{n=1}^N Y_{\pi_n,m,n}]}{\mathbb{E}[\sum_{n=1}^N \widetilde{X}_{\pi_n,n}]} \leq O(V/N), \ m = 1, 2, \ldots, K.$$

for any $N$. This will be done by showing that the `OLUM` Algorithm achieves $\epsilon$-optimal Lyapunov drift with high probability for each decision, thus achieves optimality fast as a result of the Lyapunov drift methodology. For details on Lyapunov optimization, refer to [35].

For any group $k \in [K]$, let

$$X_{k,n}^* = \min\{X_{G_n,n}, t_k^*\},$$

$$Z_{k,n}^* = \min\{X_{k,n}, t_k^*\} \sum_{m=1}^K U_m(\gamma_{m,n}),$$

$$Y_{k,m,n}^* = \min\{X_{k,n}, t_k^*\}\gamma_{m,n} - R_{m,n}(t_m^*)\mathbb{I}\{k = m\}, \ \forall m \in [K].$$

Note that these are the random variables in Lemma 1 under the optimal deadline $t_k^*$ for each group $k$.

The proof relies on a novel online learning approach based on drift-based optimization techniques. In this methodology, the dual variables $Q_n$ as defined in (6) summarize how much the constraint is violated in the past. At stage $n$, given the vector of dual variables $Q_n$, we have the drift-plus-penalty ratio (DPPR), which is defined as follows:

$$\Psi_n(k, Q_n) = -V\frac{\mathbb{E}[Z_{k,n}^*]}{\mathbb{E}[\min\{X_{k,n}, t_k^*\}]} + \sum_m Q_{m,n}\frac{\mathbb{E}[Y_{k,m,n}^*]}{\mathbb{E}[\min\{X_{k,n}, t_k^*\}]}. \tag{20}$$

The optimal algorithm therefore, aims to minimize the DPPR to achieve optimality. Let the terms in DPPR related to the auxiliary variables $\gamma_{m,n}$ be denoted as:

$$\psi_n(\gamma_n, Q_n) = \sum_{m=1}^K \big(-VU_m(\gamma_{m,n}) + Q_{m,n}\gamma_{m,n}\big). \tag{21}$$

Therefore, the DPPR can be written as follows:

$$\Psi_n(k, Q_n) = \psi_n(\gamma_n, Q_n) - Q_{k,n}\frac{\mathbb{E}[R_{k,n}(t_k^*)]}{\mathbb{E}[\min\{X_{k,n}, t_k^*\}]}. \tag{22}$$

The classical drift-based stochastic optimization techniques either assume the knowledge of the first-order moments in $\Psi_n(k, G_n)$, or they observe the outcomes for the completion of task $n$ prior to the decision. However, in online learning, since we have no prior knowledge of the mean values $\mathbb{E}[R_{m,n}(t_m^*)]$ and $\mathbb{E}[\min\{X_{k,n}, t_k^*\}]$, we define the empirical reward-per-processing-time as follows:

$$\widehat{r}_{k,n}(t) = \frac{\sum_{i=1}^{n-\tau} R_{k,i}(t)}{\sum_{i=1}^{n-\tau} \min\{X_{k,i}, t\}}. \tag{23}$$

where $n - \tau$ is the number of samples available. Similarly, let

$$r_k(t) = \frac{\mathbb{E}[R_{k,i}(t)]}{\mathbb{E}[\min\{X_{k,i}, t\}]}. \tag{24}$$

The deadline is chosen so as to maximize the reward per processing time:

$$\widehat{r}_{k,n} = \max_{t \in \mathbb{T}} \ \widehat{r}_{k,n}(t).$$

Let $\delta_k(t) = \max_{t'} \ r_k(t') - r_k(t)$ and $\delta(t) = \min_{(k,t):\delta_k(t)>0} \ \delta_k(t)$. By using Lemma 2, it can be shown that $T_n = t^*_{G_n}$ with probability at least $1 - e^{-n\Omega(\delta^2(t))}$, i.e., the optimal deadline for the chosen group $G_n$ is selected with high probability. With this deadline-selection policy, the empirical drift-plus-penalty ratio (e-DPPR) is defined as follows:

$$\widehat{\Psi}_n(k, Q_n) = \psi_n(\gamma_n, Q_n) - Q_{k,n}\widehat{r}_{k,n}. \tag{25}$$

The `OLUM` Algorithm as defined in Definition 4 is based on minimizing the e-DPPR in (25). The auxiliary variables in the `OLUM` Algorithm is chosen to maximize $\psi_n(\gamma_n, Q_n)$ over $\gamma_n$, and the group decision is independent of the choice of the auxiliary variables given $Q_n$.

The following proposition quantifies the approximation error for using the e-DPPR in the decision-making as a surrogate for the DPPR in the optimization.

**Proposition 6.** *For any given $\epsilon \in (0, \mu_*)$, we have the following inequality for the DPPR under the* `OLUM` *Algorithm:*

$$\Psi_n(G_n, Q_n) \le \min_{k \in [K]} \ \Psi_n(k, Q_n) + \frac{2\epsilon(1 + r^*)}{\mu_* - \epsilon} \sum_k Q_{k,n} + h(Q_n)O(n), \tag{26}$$

*where $\mathbb{E}[h(Q_n)] = c_1 e^{-c_2 n\epsilon^2}$ for some constants $c_1, c_2 > 0$ and*

$$r^* = \max_{(k,t)} \ \frac{\mathbb{E}[R_{k,n}(t)]}{\mathbb{E}[\min\{X_{k,n}t\}]}.$$

The proof of Proposition 6 relies on the concentration result presented in Lemma 2 and a PAC-type bound: let $k$ be a group such that $\Psi_n(k, Q_n) > \min_j \Psi_n(j, Q_n) + \delta$ for any $\delta > 0$ given $Q_n$. Then,

$$\mathbb{P}(G_n = k|Q_n) \le \mathbb{P}(|\widehat{\Psi}_n(k, Q_n) - \Psi_n(k, Q_n)| > \delta/2|Q_n)$$
$$+ \mathbb{P}(|\widehat{\Psi}_n(k_n, Q_n) - \Psi_n(k_n, Q_n)| > \delta/2|Q_n),$$

where $k_n = \arg\min_j \Psi_n(j, Q_n)$. Then, a straightforward application of Lemma 2 and union bound (over suboptimal groups) with $\delta = \epsilon \cdot O(\sum_k Q_{k,n})$ for $\epsilon > 0$ yield the result.

We have the following lemma, which will be key in the analysis of the learning algorithm.

**Lemma 4** ([35]). *Let $L(q) = \frac{1}{2}\sum_{m=1}^{K} q_m^2$ be the quadratic Lyapunov function, and*
$$\Delta(Q_n) = \mathbb{E}[L(Q_{n+1}) - L(Q_n)|Q_n],$$

*be the Lyapunov drift. Then, we have the following bound on the Lyapunov drift for the problem* (12):

$$\Delta(Q_n) \le D + \mathbb{E}\left[\sum_{k \in [K]} Q_{k,n}Y_{G_n,k,n}|Q_n\right], \tag{27}$$

*for some constant $D > 0$ under the* `OLUM` *Algorithm.*

Following the methodology in [19], from Prop. 6 and Lemma 4 with $\epsilon = \epsilon_n = \frac{2(1+r^*)}{\mu_*}\sqrt{\frac{\beta \log(n)}{n}}$ for $\beta > 2$, we have the following result:

$$\Delta(Q_n) - V\mathbb{E}[\min\{X_{G_n,n}, T_n\}\sum_k U_k(\gamma_{k,n})|Q_n] \le D$$

$$+ \mathbb{E}[\min\{X_{G_n,n}, T_n\}|Q_n]\left(-VU^* + \epsilon_n \sum_k Q_{k,n} + \mathbb{E}[h(Q_n)|Q_n]O(K \cdot n)\right). \tag{28}$$

where $D > 0$ is a constant, and the RHS holds since there exists an optimal stationary randomized policy for (12) which satisfies:

$$\min_k \Psi(k, Q_n) \le \Psi(\tilde{G}_n, Q_n) = -VU^*.$$

Taking the expectation in (28), we have:

$$\mathbb{E}[L(Q_{n+1}) - L(Q_n)] - V\mathbb{E}[\min\{X_{G_n,n}, T_n\} \sum_k U_k(\gamma_{k,n})] \leq B - VU^*\mathbb{E}[\min\{X_{G_n,n}, T_n\}]$$

$$+ \epsilon_n \max_{k \in [K]} \mathbb{E}[X_{k,n}] \sum_{k \in [K]} \mathbb{E}[Q_{k,n}] + O(K)n^{1-\beta}, \quad (29)$$

Summing the above over $n = 0, 1, \ldots, N - 1$, dividing by $N$, and rearranging terms, we have the following inequality:

$$\frac{\mathbb{E}[\sum_{n=1}^N Z_{\pi_n,n}]}{\mathbb{E}[\sum_{n=1}^N \widetilde{X}_{\pi_n,n}]} \geq U^* - \frac{2(1+r^*)}{\mu_*} O\left(\sqrt{\frac{\beta \log(N)}{N}}\right) + \frac{D/\mu_* + O(N^{\beta-2})}{V} + \frac{\mathbb{E}[L(Q_0)]}{V\mu_* N}. \quad (30)$$

The second question we had was how much the constraint in (12) is violated. From the update of the dual variables (6), we have the following:

$$Q_{k,n+1} \geq Q_{k,n} + Y_{\pi_n,k,n}, \quad (31)$$

Summing the above over all $n = 0, 1, \ldots, N - 1$, we have:

$$Q_{k,N} \geq Q_{k,0} + \sum_{n=1}^N Y_{G_n,k,n}.$$

Thus, we have:

$$\frac{\mathbb{E}[Q_{k,N}]}{N\mu_*} \geq \frac{\mathbb{E}[\sum_{n=1}^N Y_{\pi_n,k,n}]}{\mathbb{E}[\sum_{n=1}^N \min\{X_{G_n,n}, T_n\}]}. \quad (32)$$

By Lemma 3, the following inequality holds:

$$\frac{\mathbb{E}[Q_{k,N}]}{N} \leq O\left(\frac{V}{N}\right).$$

Hence, by choosing $V = \Theta(\sqrt{N/\log(N)})$, we show that the objective is achieved with $O(\sqrt{\log(N)/N})$ gap, and the constraint is satisfied at a rate $O(1/\sqrt{N\log(N)})$.

**Step 2.** In this step, we will show that the decision-making process continues for $N^\pi(B) = \Theta(B)$ stages with high probability, which will conclude the proof.

For any $B > 0$, let $n_0(B) = \lceil 2B/\mu_{min} \rceil$. Then, under any causal policy $\pi$, we have the following bound:

$$\mathrm{REG}_{\Pi_S}^\pi(B) = U^* - \frac{\mathbb{E}[\sum_{n=1}^{N^\pi(B)} Z_{\pi_n,n}]}{B},$$

$$\leq U^* - \frac{\mathbb{E}[\sum_{n=1}^{N^\pi(B)} Z_{\pi_n,n}]}{\mathbb{E}[\sum_{n=1}^{N^\pi(B)} \widetilde{X}_{\pi_n,n}]}, \quad (33)$$

$$\leq \frac{\mu^* \cdot n_0(B)}{B}\left(U^* - \frac{\mathbb{E}[\sum_{n=1}^{n_0(B)} Z_{\pi_n,n}]}{\mathbb{E}[\sum_{n=1}^{n_0(B)} \widetilde{X}_{\pi_n,n}]} + o(1)\right), \quad (34)$$

where $U^* = \mathrm{OPT}_{\Pi_S}(B) + O(1/B)$ is the optimal utility in Lemma 1, $\mu^* = \max_k \mathbb{E}[X_{k,n}]$, and (33) holds since $\sum_{n=1}^{N^\pi(B)} X_{G_n,n} \geq B$ by definition. In order to prove (34), first note that

$$\mathbb{E}\left[\sum_{n=1}^{N^\pi(B)} \left(U^*\widetilde{X}_{\pi_n,n} - Z_{\pi_n,n}\right)\right] = \mathbb{E}\left[\sum_{n=1}^\infty \left(U^*\widetilde{X}_{\pi_n,n} - Z_{\pi_n,n}\right)\mathbb{I}\{N^\pi(B) > n\}\right],$$

$$\leq \mathbb{E}\left[\sum_{n=1}^{n_0(B)} \left(U^*\widetilde{X}_{\pi_n,n} - Z_{\pi_n,n}\right)\right] + U^* \sum_{n>n_0(B)} \mathbb{P}(N^\pi(B) > n)$$

$$(35)$$

Since $\{N^\pi(B) > n\} \subset \{\sum_{i=1}^n \widetilde{X}_{\pi_i,i} < B\}$ by definition and $\mathbb{E}[\widetilde{X}_{\pi_i,i}|\mathcal{H}_i] \geq \mu_* > 0$ for all $i$, we have:

$$\mathbb{P}(N^\pi(B) > n) = \mathbb{P}(\sum_{i=1}^n \widetilde{X}_{\pi_i,i} < B) \leq e^{-n\Omega(1)},$$

for all $n > n_0(B)$ by Azuma-Hoeffding inequality [53], which implies that $n_0(B)$ is a high-probability upper bound for $N^\pi(B)$ under any causal policy $\pi$. In other words, the decision-making process continues for at most $n_0(B)$ turns with high probability since each action depletes a positive amount from the time budget $B$. Consequently, we have

$$\sum_{n>n_0(B)} \mathbb{P}(N^\pi(B) > n) \leq e^{-\Omega(B)} = o(1).$$

Using this result and rearranging the terms in (35), we obtain the inequality in (34). Furthermore, the constraints are satisfied at rate $O(V/B)$ for all groups. Therefore, by using the result of Step 1 with $N = n_0(B)$ and noting that $n_0(B)/B = \Theta(1)$, we conclude that $\mathtt{REG}_{\Pi_S}^\pi(B) = O(\sqrt{\log(B)/B})$.