[Reviews · NeurIPS 2020]

Review 1

Summary and Contributions: The paper addresses an online learning problem with group fairness and task completion time constraint. The paper proves approximation results for the optimal offline policy and proposes an algorithm OLUM for online setup. They also prove that it achieves \sqrt{log B/B} regret bound for a given time budget B.

Strengths: 1. The problem statement has practical implications in real-life problems such as hiring in online freelancing platforms and server allocation in cloud computing. This problem is less studied in existing literature. 2. The result of fair resource allocation in continuous time is novel and interesting. Specifically its reducibility to multiple utility functions is useful. 3. The OLUM algorithm and its analysis for this problem setup is useful and can be extended for future works. 4. Motivation of the problem is well defined.

Weaknesses: 1. The motivation takes too much of the paper and interesting methodological contributions such as elaborations of the concepts in Sec 3 is too brief. This section consists of too many concepts without enough intuition. 2. The assumptions like i.i.d returns across tasks and group-dependency should be discussed further. 3. The proof techniques should be discussed briefly in the paper. Specifically, the hardness of continuous time problem, Lyapunouv drift etc would be interesting to a theoretically inclined reader. 4. The empirical results are too primary with respect to the motivation. Some work on real data for one of the examples would satisfy the expectation built-up in introduction.

Correctness: The results seem correct in the given setups.

Clarity: 1. The paper is well motivated. 2. The language is clear and fluid. 3. The concepts in sec 3 need further elaboration for better understanding. 4. The proof sketch and the hardness of the setup should be discussed.

Relation to Prior Work: 1. The paper has healthy amount of references. 2. Contrasts and improvement of results from [19] should be discussed further.

Reproducibility: Yes

Additional Feedback: 1. Why assuming group-dependent and i.i.d tasks for rewards? This seems like a bit simplistic as different people in the same group (eg demography) can have different skillsets and rewards. Justifying this assumption and discussing the limitations would be crucial. 2. Too much space is spent on motivation which is well-written but no empirical result or theoretical result is provided to satisfy this. It would be interesting to validate the effectiveness of the proposed solutions for at least one of the problems. 3. Use of \alpha-fair allocation is general and interesting. Cor 1 illustrates the optimal utilities for proportional fairness and reward maximization. Similarly, deriving solutions for max-min and min delay fairness would be useful. Authors may like to put a table to for the results for different \alpha's. Then put more emphasis on elaborating the intuitions. 4. Contrasts and improvement of results from [19] should be discussed further. 5. The proof techniques should be discussed briefly in the paper. Specifically, the hardness of continuous time problem, Lyapunouv drift etc would be interesting to a theoretically inclined reader.


Review 2

Summary and Contributions: This paper considers an online task allocation problem, with budget and fairness constraints, in a continuous-time setting.

Strengths: The problem presented in this paper is compelling certainly relevant. The results seem plausible and correct, though I was not able to fully understand their model (see below).

Weaknesses: The model formulation is very confusing. For example, the notion of "feedback" is apparently pretty important to this model--however the notion of "feedback" is not described before it's attached to a mathematical quantity H_k, n-1. Furthermore, it's not clear what kind of mathematical object H is. I would encourage the authors to re-structure their section introducing their formalism, and pass it by researchers not familiar with this project--this might help improve the clarity. This entire section was difficult to follow, even after reading through it several times. Some specific points of confusion (perhaps due to my not fully understanding the formalism): - how are tasks assigned to individuals within a group? is this important? - why is there a notion of "groups", and why not just "individuals"? since there seems to be no formalism describing individuals, it seems that groups are not *sets* of servers or agents, but rather groups *are* the agents. If my understanding is correct, then it's somewhat dubious to claim that this paper considers group fairness. If my understanding is incorrect, then the authors should please please clarify the model and formalism. - is the goal of your allocation policy to learn? or is it to allocate resources efficienctly? Or both, as in some bandit settings? Correctness: Are the claims and method correct? Is the empirical methodology correct? * (visible to authors during feedback, visible to authors after notification, visible to other reviewers, visible to meta-reviewers) Not sure. Since I wasn't able to fully understand the model, I could not fully assess the authors claims.

Correctness: The paper is readable, but several aspects of the model and notation are very confusing; I didn't check the theoretical claims.

Clarity: The paper is readable, but several aspects of the model and notation are very confusing.

Relation to Prior Work: The related work section is understandable.

Reproducibility: Yes

Additional Feedback:


Review 3

Summary and Contributions: This paper considers the problem of online allocation problems in known and unknown distributions with goal of using allocation algorithms that are "group-fair". Mathematically, this is represented by a utility function for each group that maps the average reward rate for this group to a real number and is concave and monotone. The goal of the algorithm is to maximize the sum of utilities across the $k$ groups. Given a policy class $\Pi$, the goal of the algorithm is to minimize regret with respect to the maximum utility obtained by any policy in this class. In particular, at each time-step, the algorithm needs to choose the group it assigns the task to and the deadline. If the task is not completed within the specified deadline, the algorithm does not get any reward. The paper first provides a characterization of the problem in the known distribution/offline setting. In particular, they show that there exists a fixed distribution $P^*$ over the space of groups and time-steps, such that playing at each time-step sampling a decision using $P^*$ is asymptotically optimal. This is akin to the various solution concepts used in the literature of online learning and online algorithms, where one characterizes the upper-bound on the optimal value using a fixed distribution. Then they propose their main online learning algorithm. This is based on maintaining empirical estimates of the delays and the rewards. They maximize a scaled function of the reward to delay ratio, where this scaling is chosen using lagrange duality. However, this algorithm makes the key assumption of full-feedback with delay. In other words after a specififed \tau time-steps, the algorithm sees the reward of all the groups.

Strengths: The strengths of this paper are as follows: (1) The paper considers an important problem of group-fairness in online allocation problems. Group fairness is an important aspect and has seen a lot of interest in both the ML community and the larger AI/CS community. Moreover, considering fairness in the online learning setup is novel and thus, highly relevant to the conference. (2) The paper makes justified assumptions on the nature of the utility function. Moreover, the considered utility function family captures _all_ the commonly considered notions of fairness. Thus, the theoretical results in this paper could fairly be well-translated to the claimed applications (upto a small caveat, which I elaborate below in the weakness). (3) Clean algorithm: The other strength of this paper is that the algorithm is pretty clean from a practical implementation standpoint. Of course, one needs access to the inverse of the derivative of the utility function. But for the $\alpha$-fair family this is fairly easy to compute.

Weaknesses: (1) The first weakness of this paper is that the online learning algorithm makes an assumption about full-feedback after a certain time-delay. This seems unreasonable to me in the applications, since it is a-priori unclear how you would get the quality estimates for other groups given a task. The paper starts with a pretty general model where the tasks across different time-steps can be pretty heterogenous. So isn't then the case that getting counterfactual estimates for other groups unreasonable? Is this assumption so that the technical analysis goes through? In that case, please mention that. Also discuss why it may be okay if the tasks are similar. This is my main worry about the significance of the results in this paper. (2) The other weakness of this paper I found was that it was pretty notation heavy to get through. In my opinion, the notations can either be simplified and/or be accompanied by simpler easier to read english statements. For example, Prop. 2, 4, 5 have a quantities with numerator and denominator and bounds are being presented with respect to this quantities (the one with equation line in latex). However, these quantities are not so easy to read and interpret immediately. As a presentation, you could explain in simple terms what these mean and why comparing to these bounds is useful? For instance in Prop 4, the \phi_k just implies the prob. of delay for group k. (Also, I think there is a typo. It is defined using \phi_k but the bounds are on rho_k). (3) Finally, something seems odd in the theorem 1 (is it hiding the term R_max)? Usually regret bounds also scale with how the R_max scales. Since the model does not place any bounds on R_max, it may be good to not hide it within the O(.) notation.

Correctness: Overall, the theorem statements seem sound and I could not find much discrepencies (except the minor comment above). I have not checked the proofs in the supplementary materials thoroughly.

Clarity: It is mostly well-written but could do better. See comments above.

Relation to Prior Work: Yes, the paper clearly describes the relationship to prior work and supplementary materials has a detailed list of related work.

Reproducibility: Yes

Additional Feedback: After rebuttal, here are my comments. Thanks for the detailed response to the reviewers questions, especially in such a short period of time! It is good to see that the authors will put the effort to improve the presentation! I think my other weakness I stated was about the application; in particular the full-feedback model with delay. Essentially the claim is that after a few days, the platform will share performance about other workers ( in two-sided crowdsourcing market setting). However, this estimate can at-best be noisy and has to be learnt from other tasks this worker has performed and transfer that to feedback about this task. So assuming exact feedback after delay still seems like a stretch to me, in practice. In general, I think the theory in the paper is pretty neat and the contributions are novel. I would really like to see a bit of discussion/caveats added in the intro where the authors elaborate that they need to make some simplifying assumptions that may not hold in practice but is required for the technical analysis to go through. With this in place and the other presentation improvements the authors promised, this paper would be reasonable for acceptance.


Review 4

Summary and Contributions: The paper considers a continuous-time online learning problem with fairness considerations, and devises a framework based on continuous-time utility maximization. By using general enough utility functions, the framework can recover many special cases such as reward-maximizing, max-min fair, and proportionally fair allocation across different groups. The authors characterize the optimal offline policy, and subsequently introduce an online learning algorithm which achieves a guaranteed regret bound.

Strengths: 1. This is a solid technical work which provides (1) characterization for optimal policy as well as (2) an online algorithm with a guaranteed regret bound. The use of renewal theory is interesting. 2. The utility function is broad enough to encompass many different cases, including reward-maximization and proportional fairness. 3. Several motivating examples.

Weaknesses: 1. It is nice that there is an experimental section, but it refers to a very simple setting. It would have been more convincing if the authors included results with more than two groups (even in the appendix). 2. The feedback model of Section 4 seems to be too strong. In the online case, if we choose one specific group for task n, how can we expect to have delayed information over all groups for task n? Of course, by making this assumption, the authors can get good empirical estimates, so such an assumption seems to be convenient to complete the proof. But despite the motivating examples behind it, I do not find it very realistic.

Correctness: I managed to proofread only few of the claims/proofs, and they were correct. The other claims seem to be correct, but I am not fully certain.

Clarity: The paper is not too hard to follow, even though the main text perhaps contains too many technical details (which could have gone to the appendix, and leave space in the main text for more discussion, high-level comments and explanations etc.).

Relation to Prior Work: Prior work is covered well, and there is discussion on how this work differs from previous ones.

Reproducibility: Yes

Additional Feedback: 1. Regarding point 2 in the weaknesses section above, could you have used a different feedback model that does not assume full delayed information over all groups for task n? 2. Have the authors performed experiments with more than two groups? How does the online algorithm behave under the different regimes of the utility function (e.g., in terms of convergence or regret)? The setting with 2 groups is encouraging but quite simple. ---UPDATE--- I thank the authors for the detailed rebuttal. I suggest they include their clarification on the full delayed information model as well as more experiments in the updated version, in accordance with the author response.

[Author Response · NeurIPS 2020]

We thank the reviewers sincerely for their valuable feedback, which has provided us with an achievable plan to
improve the clarity and impact of the paper with a revision as outlined below. We were particularly encouraged by the
predominantly positive and supportive remarks (e.g., R1: "*The result of fair resource allocation in continuous time is*
*novel and interesting.*" and R4: "*This is a solid technical work which provides (1) characterization for optimal policy*
*as well as (2) an online algorithm with a guaranteed regret bound. The use of renewal theory is interesting.*") In the
following, we present our responses to the specific questions of the reviewers.

**Q1.  Clarification of the system model and objective (R2)** We would like to thank Reviewer 2 for pointing this
out. We will enhance the presentation of the system model to avoid confusions and improve clarity. We consider a
controller (or decision maker) that allocates tasks sequentially to one of $K$ groups. A group is analogous to an *arm* in
the stochastic bandit setting, where arm pulls from a given arm yield i.i.d. rewards. Analogously, each task completion
(by an individual) from a given group yields a random reward. We assume these completion times and rewards are i.i.d.
within each group. The decision-making process continues until the total time spent exceeds a given time budget $B$,
and the objective of the controller is to maximize the utility subject to this budget constraint. Using different utility
functions, we are able to learn a fair allocation of the budget $B$ across groups with different notions of fairness, e.g.,
proportional fairness. Since the controller does not have any prior statistical knowledge on the completion times and
rewards for any groups, learning and utility maximization need to be performed simultaneously. Our i.i.d. [reward,
completion time] assumption implies a statistical symmetry between individuals (*arm pulls*) within a group (*arm*).
Therefore, allocation of individuals within groups is *not* a part of the decision-making in our setting.

**Q2. Why do we assume group-dependent and i.i.d [reward, completion time] pairs? (R1, R2)** We would like to
thank the reviewers for this important question. We will further justify the statistical assumptions in the final version
of the paper. In many cases, the individuals within a group exhibit random but *statistically similar* (i.e., independent
and identically distributed) performance as a consequence of their common demographic (e.g., resources for training)
background. We agree that individuals within a given group may have very different skill sets, and we reflect this as
random and potentially *heavy-tailed* completion times for each individual, which implies that there can be frequent
statistical outliers within a group that deviate from the mean tremendously depending on the tail. We also note that our
approach in this paper can be utilized to develop algorithms for different and more complicated statistical models, such
as time-varying (e.g., Markov-modulated) completion time and reward statistics.

**Q3. Why do we assume delayed full information feedback? (R3, R4)** As noted by Reviewer 1, a key contribution
of this work is to introduce the powerful Lyapunov drift methods into the fair online learning framework. This is a
challenging task because of the interaction between the state dynamics and the empirical estimates. Thus, as Reviewer
3 points out, the delayed full feedback assumption facilitates the analysis, and provides a methodological basis for other
more complicated feedback models (analogous to the *expert advice* in bandits). We will further justify in the paper that
there are applications for which the delayed full-feedback model is accurate. For example, in server allocation and
two-sided matching markets with online reviews, the completion time and reward realizations (or quality estimates
through interpolation) are shared among decision-makers after a delay. Thus, the feedback of an unchosen group for
the $n$-th task is available only after $\tau$ tasks. Nevertheless, we agree that there are also many applications for which
full-information feedback is unavailable, which we leave as an open question. We believe this question is highly
non-trivial since state-dependent online exploration will be crucial in partial feedback models, which makes the analysis
of the Lyapunov drift methodology considerably harder because of the complex interactions between the state and
statistical estimate for each decision.

**Q4. Extended Performance Evaluations of the `OLUM` Algorithm (R1, R4)** In the final version of the paper, we will
include extended numerical investigations. Our updated set of experiments will address the motivating examples of
server allocation in cloud computing systems and contractual hiring in online freelancing platforms, for which accurate
power-law statistical models are available in the literature (e.g., [36], [37]) , and include: (i) the impact of the alpha
fairness parameter on the performance of the `OLUM` Algorithm, (ii) the case of multiple groups (as suggested by Reviewer
4), and (iii) the impact of the tail distributions on the resource allocation. By these investigations, we will observe how
regret scales with the number of groups and statistical heterogeneity in practice.

**Q5. Writing and Clarifications (R1, R3)** We will expand the technical sections in the paper to provide more insight
on the theoretical novelties that we introduce. In particular, we will emphasize the novel challenges and our solutions in
developing a learning algorithm based on empirical Lyapunov-drift estimates as suggested by Reviewer 1, this will also
highlight the difference with [19], which uses an exact Lyapunov drift that requires complete statistical knowledge.
Also, we will add more insights on our renewal theory-based approach to utility maximization in continuous time (as
suggested by R1), which is a less-studied problem in the literature. We will also clarify the impact of $R_{max}$ in the
theorem statement, which was previously discussed in the proof (as mentioned by R3).

[Meta-Review · NeurIPS 2020]

The submission had four reviewers. The scores of the original reviews were somewhat varied. The reviewers considered the problem setting generally well motivated and interesting, and the authors make solid progress with interesting results. However, questions were raised about the appropriateness of some particular assumptions of the model. There is some empirical evaluation, which is good, but in its present stage it's much too limited. All the reviewers also pointed out shortcomings in the presentation, some of them serious. The authors' reply addressed the questions about the model and satisfied the reviewers partly, but not entirely. The authors also clarified some unclear points in the current version and promised more work on the presentation in a revised version, and more extensive experiments. As a result of the authors' reply and lively discussion among the reviewers, some of the reviewers raised their evaluation, and they are now unanimously proposing accepting the paper. However, some reservations remain. Overall, I consider the submission somewhat above the acceptance threshold.